# Prediction of AI-Based Personal Thermal Comfort in a Car Using Machine-Learning Algorithm

Yeong Jo Ju [1] , Jeong Ran Lim [2] and Euy Sik Jeon [2,3,*]

[1] Graduate School of Mechanical Engineering, Kongju National University, Cheonan-daero, Seobuk-gu, Cheonan-si 31080, Korea; robos10@kongju.ac.kr

[2] Industrial Technology Research Institute, Kongju National University, Cheonan-daero, Seobuk-gu, Cheonan-si 31080, Korea; jeongran@kongju.ac.kr

[3] Department of Future Convergence Engineering, Kongju National University, Cheonan-daero, Seobuk-gu, Cheonan-si 31080, Korea

* Correspondence: osjun@kongju.ac.kr

**Abstract:** Defining a passenger's thermal comfort in a car cabin is difficult because of the narrow environment and various parameters. Although passenger comfort is predicted using a thermal-comfort scale in the overall cabin or a local area, the scale's range of passenger comfort may differ owing to psychological factors and individual preferences. Among the many factors affecting such comfort levels, the temperature of the seat is one of the direct and significant environmental factors. Therefore, it is necessary to predict the cabin environment and seat-related personal thermal comfort. Accordingly, machine learning is used in this research to predict whether a passenger's seat-heating-operation pattern can be predicted in a winter environment. The experiment measures the ambient factor and collects data on passenger heating-operation patterns using a device in an actual winter environment. The temperature is set as the input parameter in the measured data and the operation pattern is used as the output parameter. Based on the parameters, the predictive accuracy of the heating-operation pattern is investigated using machine learning. The algorithms used in the machine-learning train are Tree, SVM, and kNN. In addition, the predictive accuracy is tested using SVM and kNN, which shows a high validation accuracy based on the prediction results of the algorithm. In this research, the parameters predicting the personal thermal comfort of three passengers are investigated as a combination of input parameters, according to the passengers. As a result, the predictive accuracy of the operation pattern according to the tested input parameter is 0.96, showing the highest accuracy. Considering each passenger, the predictive accuracy has a maximum deviation of 30%. However, we verify that it indicates the level of accuracy in predicting a passenger's heating-operation pattern. Accordingly, the possibility of operating a heating seat without a switch operation is confirmed through machine learning. The primary-stage research result reveals whether it is possible to predict objective personal thermal comfort using the passenger seat's heating-operation pattern. Based on the results of this research, it is expected to be utilized for system construction based on the AI prediction of operation patterns according to the passenger through machine learning.

**Keywords:** personal thermal comfort; heating seat; heating operation pattern; cabin environment; classification algorithms

## 1. Introduction

The heating, ventilation, and air-conditioning (HVAC) system of a vehicle provides thermal comfort to passengers in various cabin environments. The cabin inside the vehicle is a highly asymmetrical environment. Thus, factors, such as cabin temperature, air velocity, and air humidity around the passenger, play an essential role in determining passenger thermal comfort [1]. Recently, as vehicles have been converted into high-tech

autonomous vehicles, such as hybrid electric vehicles and electric vehicles (EVs), research on an efficient method to maintain the balance of energy saving and thermal comfort has gained importance [2]. In conventional vehicles, heat from the engine is used for the HVAC of the cabin; however, in the case of EVs, the cabin's thermal load consumes approximately 1/3 of the stored electricity. In practice, if there is not enough energy to satisfy the passenger's thermal comfort, the use of the HVAC system should be reduced to reach the final destination or next charging point. In some situations, the speed must be reduced to satisfy the passenger's thermal comfort [3].

Research on thermal comfort has mainly been conducted in the field of buildings. Most approaches involved the design of HVAC systems to provide adequate thermal comfort for large groups of building occupants. Although the average response of the occupant group was focused on earlier, the focus recently shifted to predicting individual thermal-comfort responses [4]. It assumes that considering thermal-comfort sensitivity affects the overall probability of achieving comfort. A study was conducted using this approach to investigate the effect of individual thermal-comfort sensitivity (individual response to temperature change) on collective conditioning [5].

Research on personal comfort for seat HVAC (auxiliary air condition) systems starts in the main HVAC of the indoor and vehicle environments. The HVAC system focused on the overall thermal comfort of the passengers in the cabin and the research was mainly conducted on it. For thermal comfort in the cabin, a study was conducted to estimate the thermal sensation or sensitivity according to the outdoor climate or cabin climate using a virtual thermal model and a human subject. The thermal-comfort level of the occupant was evaluated in a non-uniform thermal environment in the vehicle interior using a numerical, analytical comfort model [6]. The thermal-comfort model for the cabin environment was verified using data from human subjects. Sensitivity studies of various cabin environments were conducted using an analytical method [7], including the discharge temperature, respiration-level temperature, and wind speed. For the ventilation system, the overall thermal-comfort and sensation levels applicable were predicted using the computational fluid dynamics (CFDs) model. The predicted results were compared with the human-subject test. [8]. Through the CFD model of the virtual thermal model inside the vehicle, the sensitivity of the thermal sensation equivalent to that of the actual passenger for various parameters, such as various HVAC systems, was confirmed [9]. Moreover, it was confirmed that there is a high correlation between the actual test data and model simulation results [10]. The studies were based on thermal comfort according to cabin air conditions in the passenger and the surrounding spaces. In addition, based on the comfort index of the OEM, research was conducted mainly on the verification of human-subject data.

The cabin's air-conditioning system aims to make the passengers thermally comfortable. However, a passenger's thermal comfort is affected by many environmental variables. The thermal-comfort preference can vary significantly individual-wise, due to the physiological and behavioral factors. Such variance makes it necessary to predict personal thermal comfort and the functioning of the seat's auxiliary HVAC system [11]. Studies were conducted on thermal comfort using AI in an indoor building environment as well a vehicle environment. One of them was directed to predict the thermal comfort of occupants in a building's daily living environment or an office environment. A study tried to predict the individual thermal comfort using the Internet of Things (IoT) and machine learning from data collected from multiple or individual daily environments [12,13]. To accurately predict the thermal-comfort index, a supervised learning machine, providing output samples in the learning phase, was applied. SVM machine learning was performed to predict representative experimental factors: temperature, average radiant temperature, relative humidity, air velocity metabolism, and clothing values that affected human heat balance [14]. To estimate the temperature of passenger satisfaction from the sensors in the vehicle's interior, a study was conducted using several machine-learning methods. The passenger's experimental temperature and indoor data were collected through testing under various environmental conditions, and a study was conducted to validate the evaluation implemented by the

machine-learning approach, for data validation [15]. By combining the simulation and machine-learning algorithms, the thermal comfort of the vehicle passenger was predicted to combine the flow characteristics for the environmental conditions inside the vehicle and the HVAC setting. A cabin model validated for experimental measurements was used to generate the boundary conditions affecting the passenger's thermal comfort. Another study was conducted to predict the thermal comfort by applying a machine-learning algorithm to the simulation, considering the cabin's environmental conditions and HVAC settings [16].

In this study, the thermal comfort of the occupants and passengers according to the indoor and vehicle environments is predicted. The main objective of these studies is to predict the thermal comfort of multiple occupants or passengers for HVAC in general. Although studies have targeted individuals rather than the overall or local areas, the research on predicting personal thermal comfort for individual auxiliary air-conditioning systems is insufficient. The approach to predict personal thermal comfort used a subjective thermal-comfort vote; however, owing to the complex environment of the cabin, the prediction of personal thermal comfort may not agree with the comfort scale. Research predicting thermal comfort based on the passenger's pattern, or approach, is insufficient. Accordingly, a novel approach to predict thermal comfort is required. Therefore, it is necessary to study whether auxiliary air conditions can be operated switch-free, by learning the operation patterns of each passenger. The gap between this and existing research is prediction and confirmation of the possibility of personal thermal comfort using the operation-pattern data measured by direct contact with the passenger (seat temperature) for an individual seat air-conditioning system. In existing research, it was predicted through a passenger's thermal comfort vote; however, in this research, it can be seen that the difference is in predicting objective thermal comfort with an approach using a thermal operation pattern.

In this research, various operation patterns of passengers for vehicle heating seats are predicted using machine learning. The possibility of pattern learning is confirmed. For this, ambient and passenger data are measured. The temperature data of the outdoor, the cabin, the seat, and measurement data according to heating operation are set as the input data for machine learning. The training and test datasets are created through data normalization. Using the dataset, we derive prediction accuracy (AUC) through the training of three algorithms, and decide on two algorithms for testing. Subsequently, the predictive accuracy is confirmed according to the dataset. The high prediction results through machine-learning algorithms confirm the possibility of operating the heating seat without a switch operation. This result can be viewed as a basic stage of research on whether it is possible to predict objective personal thermal comfort using the seat-heating operation pattern measured for each passenger. Section 2 presents the setup of a recording device, hereafter referred to as a multifunctional measurement device (MFMSD). It is for measuring ambient variables and heating-operation patterns through actual vehicle driving in winter. In Section 2, the train and test datasets are created for use in machine learning. The prediction results of the passenger heating pattern using the three algorithms are presented. In Section 4, we discuss the input variables for predicting seat-heating patterns and the possibility of pattern learning. A summary of the research and future research directions are presented in Section 5.

## 2. Experiments

### 2.1. Multifunctional Measurement Device (MFMSD)

An MFMSD was used to acquire data on the seat temperatures of the driver and passengers during vehicle operation. The ambient environment data were set to collect passenger heating-operation-pattern data for a heating seat. The device was designed so that long-term recording was possible when the vehicle was in an ON/OFF state, and it could be recharged through vehicle power or operated independently using an auxiliary battery. Figure 1 presents a schematic of the measurement sensors attached to the MFMSD. They include a sensor for temperature measurement of the seat (analog devices,

TMP36), a module for temperature and humidity measurement of the cabin and outdoor (AOSONG, AM2301), a sensor for current and voltage measurement (FD14L), control, and chipset module (CH341SER). The experimental device was configured to collect ambient environment data (outside temperature, cabin temperature, and cabin humidity), passenger data (face-skin temperature), and seat-heating data (seatback temperature, seat-cushion temperature, voltage, and current) at all times. It was equipped with a status diagnosis-and-display function.

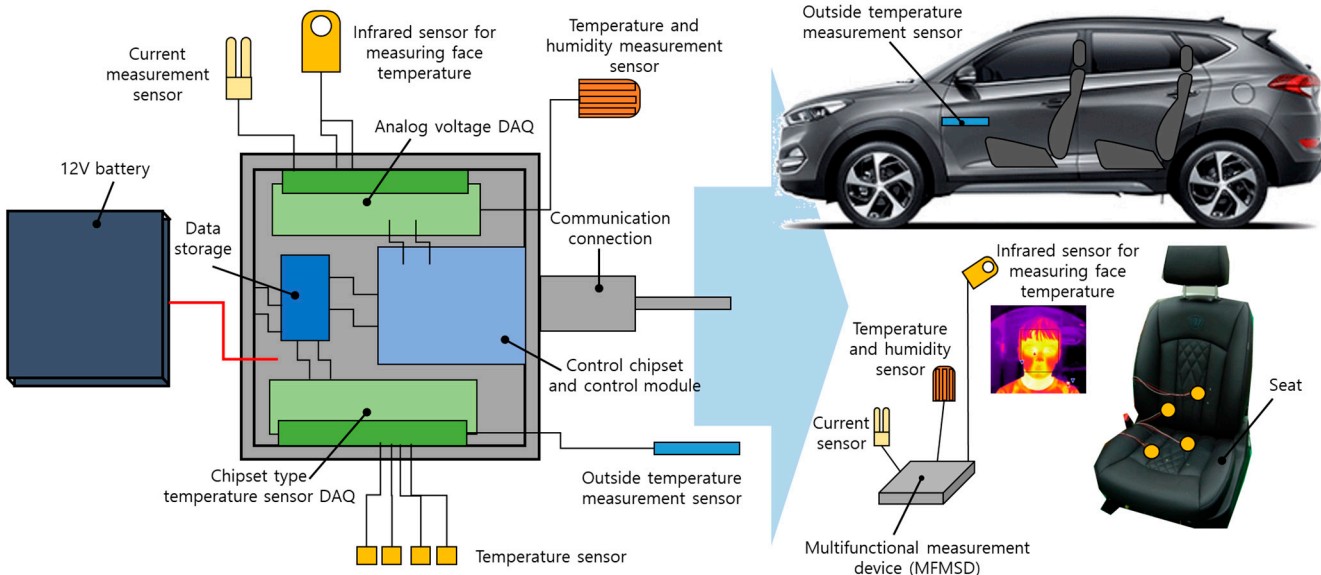

**Figure 1.** Schematic diagram of equipment setup.

The sensor and storage sets were confirmed by connecting to a PC or tablet through wired communication during the initial setting (regular inspection) to confirm the operational status of the MFMSD. Before vehicle driving, a temperature-deviation test of the temperature sensor was performed using the MFMSD for device operation in a laboratory environment. A temperature sensor was attached to two types of vehicle seats to which the device was to be attached, and a test was performed according to the temperature change of the seat for 15 min. It was confirmed that there was no significant difference within 1 °C deviation of all temperature sensors by laboratory tests.

After the MFMSD was confirmed to be operating normally, an experiment was conducted by mounting it on two real-life vehicles (vehicle 1 and vehicle 2). As shown in Figure 2, the MFMSD was used to attach temperature sensors to the seats and infrared thermometers to vehicles 1 and 2. An infrared thermometer was used to measure the face-skin temperature of the passenger during the vehicle being driven. Vehicle 1 was equipped with a seat-heating function by default. Hence, a measurement device was installed to measure the seatback temperature, cushion temperature, ambient temperature, and humidity. Vehicle 2 was not equipped with a seat-heating function. Thus, a heating seat was remodeled and installed in the driver's seat to collect the seat temperature, ambient temperature, and humidity data. For the overall air condition of the cabin, the heater was operated at a constant temperature during initial driving, and then maintained without any other adjustments.

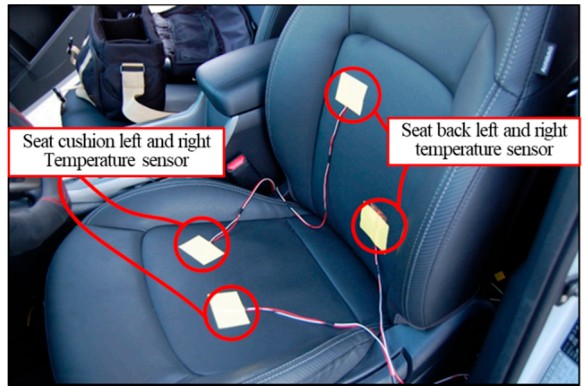

(**a**) Seat's left and right temperature sensor

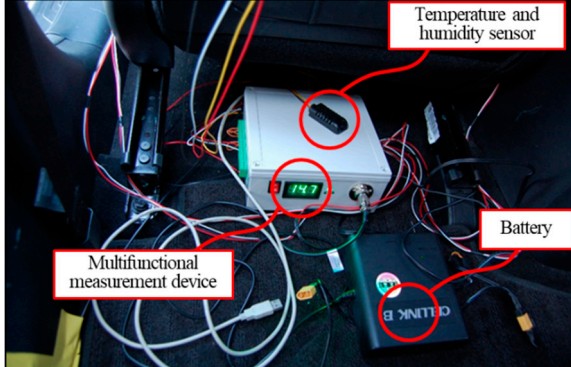

(**b**) Multifunctional measurement device

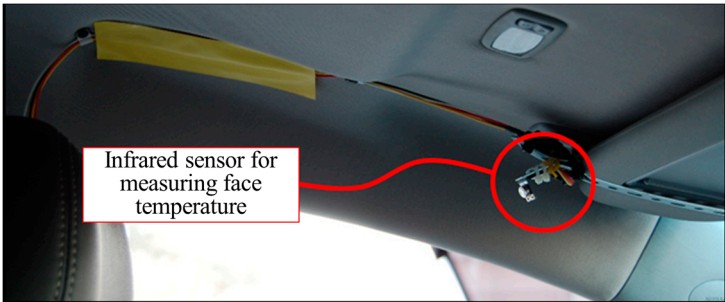

(**c**) Contactless temperature sensor

**Figure 2.** MFMSD installation in actual vehicles: (**a**) seat's left and right temperature sensor; (**b**) multifunctional measurement device; and (**c**) infrared sensor for measuring face-skin temperature.

### 2.2. Measurement

To confirm the heating-operation patterns of the passengers, measurements were made for three passengers. An experiment considering real-life vehicle driving in winter was conducted by driving approximately 310 km with two vehicles, as shown in Figure 3. Three passengers and two vehicles performed experiments at different times and measured the driving records for each vehicle, as shown in Table 1. The MFMSD was installed in the driver's seat, and data on the seat-heating use were collected when the vehicle moved for more than 40 min for 28 days. The purpose and method of collecting the seat-heating-operation pattern data were explained to the passengers. Moreover, they were asked to switch their seat-heating on or off while the vehicle was being driven. In addition to the data collected and stored by the MFMSD, the passengers were asked to note any anomalies that occurred during data recording in an experiment log. It was used as a reference for the analysis.

**Table 1.** Information on the vehicles and passengers for actual driving.

| Passenger | Gender | Test Vehicle | Distance of Driving |
|:---:|:---:|:---:|:---:|
| A | M | Vehicle 1 | 320.14 km |
| B | M | Vehicle 2 | 311.30 km |
| C | M | Vehicle 2 | 295.81 km |

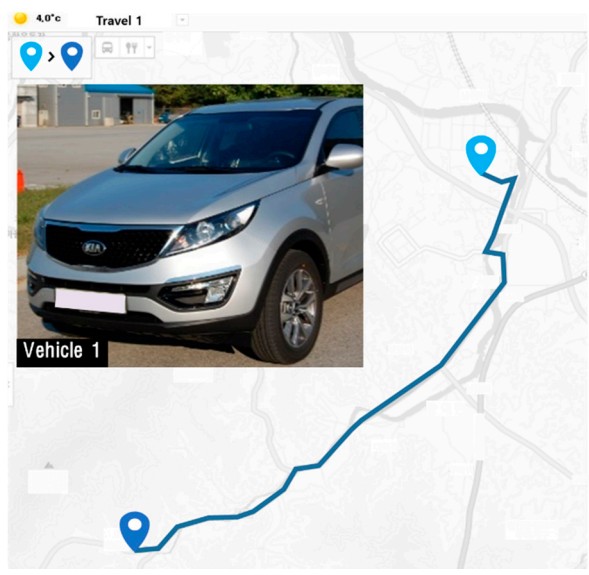
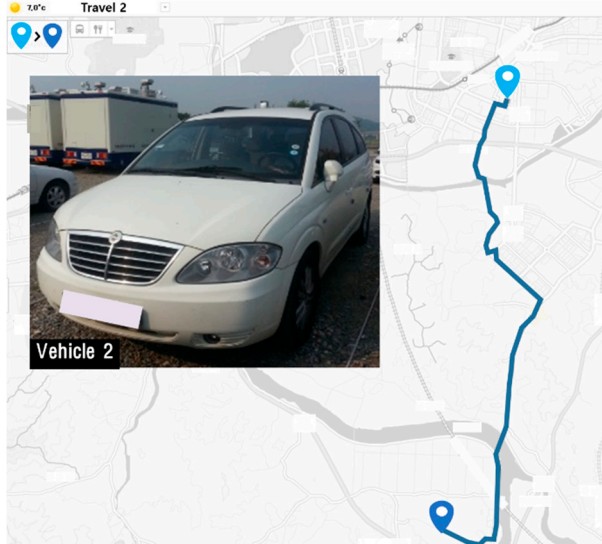

(**a**) Vehicle 1 and driving path          (**b**) Vehicle 2 and driving path

**Figure 3.** Driving route of the actual vehicle: (**a**) vehicle 1 and (**b**) vehicle 2.

As shown in Table 2, the data collected in the recording and storage device of the experimental device include the outdoor temperature, cabin temperature, cabin humidity, seatback left temperature, seatback right temperature, seat-cushion left temperature, seat-cushion right temperature, passenger's face-skin temperature, direct current (DC) voltage, and (DC) current.

**Table 2.** Data recorded during actual driving.

| Symbol | Description |
|---|---|
| Outdoor Temp | The temperature outside the vehicle measured while driving |
| Cabin Temp | The temperature inside the vehicle measured while driving |
| Cabin Humid | Humidity inside car measured while driving |
| Back Left Temp | The temperature on the left side of the seat back measured while driving |
| Back Right Temp | The temperature on the right side of the seat back measured while driving |
| Cushion Left Temp | The temperature on the left side of the seat cushion measured while driving |
| Cushion Right Temp | The temperature on the right side of the seat back measured while driving |
| Face Temp | Face-skin temperature of passenger measured while driving |
| (DC) Voltage | DC voltage while driving |
| (DC) Current | DC while driving |

The data presented in Table 2 were processed. The data recorded when the ON/OFF operation of the heating seat was not operated were excluded. All passenger data during the experiment were combined into one file. After excluding the part with abnormal signs, it was converted into a single file by referring to the experimental log.

Although the measurement data of the heating seat differed depending on the passenger, approximately 24 data sets were collected. Among them, it was confirmed that the recording time corresponding to approximately 4 h was valid. We invalidated the data similar to operation pattern of the heating seat based on the outdoor temperature,

cabin temperature, and seat temperature for each passenger. Using these data, the cabin temperature, face-skin temperature, seat temperature, and current (DC) were confirmed. The relationship between the temperature and use-pattern was confirmed by assuming the DC of the ON/OFF. As shown in Figure 4, the cabin temperature according to the passenger constantly increased regardless of the outdoor temperature and the operation of the heating seat during driving. There is a section in which the cabin temperature is kept constant without rising; however, the overall trend rises to approximately 65% of the total rise temperature, from the initial driving period to approximately 7 min, and then rises with a small deviation. The face-skin temperature of the passenger increased with the same temperature deviation as the cabin temperature increased. In addition, the humidity of the cabin increased up to 45% in the section where the temperature was greatly increased, and then decreased to a maximum of 50%.

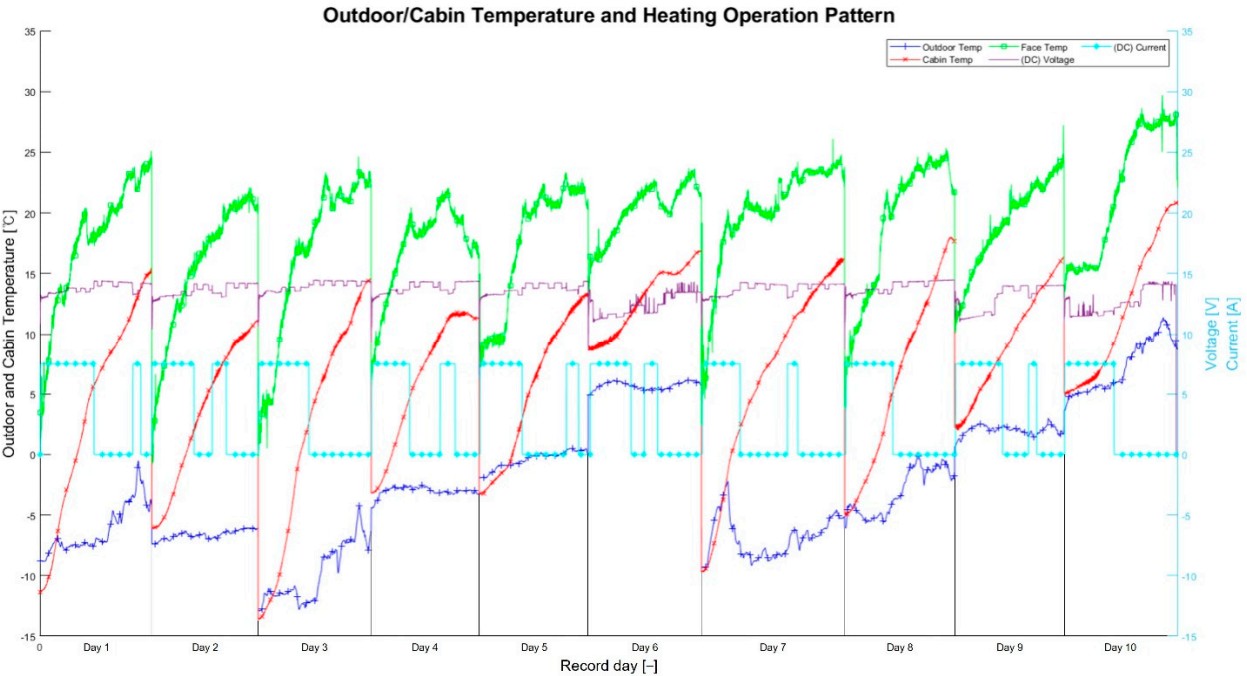

**Figure 4.** Outdoor and cabin temperature of a vehicle.

However, when the heating seat was turned ON/OFF while driving, it was confirmed that the cabin temperature was related to the overall air-conditioning system. In the case of the seat temperature, as shown in Figure 5, it was confirmed that the temperature rises and falls at the exact location according to the seat's heating ON/OFF pattern, unlike the cabin temperature. When the heating was operated at the initial driving stage, the temperature increased rapidly with a significant deviation. In the section where heating was operated, the temperature of the seat increased and decreased from the point when the operation was stopped. It was confirmed that the temperature of the seat back was higher than the seat-cushion temperature. It was also confirmed that the operation pattern of the heating seat according to the passenger was different at the same measurement time. As shown in Figure 6, passengers A and C operated the heating seat more frequently than passenger B during the same period, and, accordingly, it was confirmed that the temperature change of the heating seat was high.

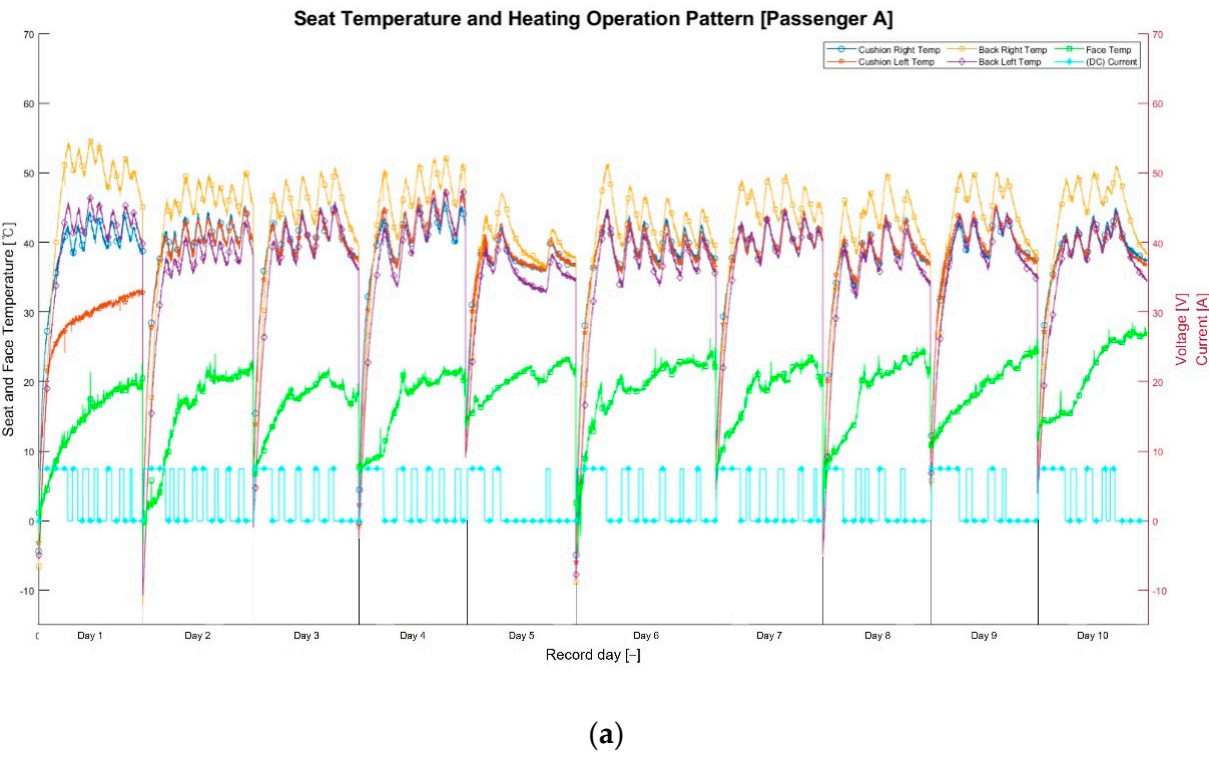

(**a**)

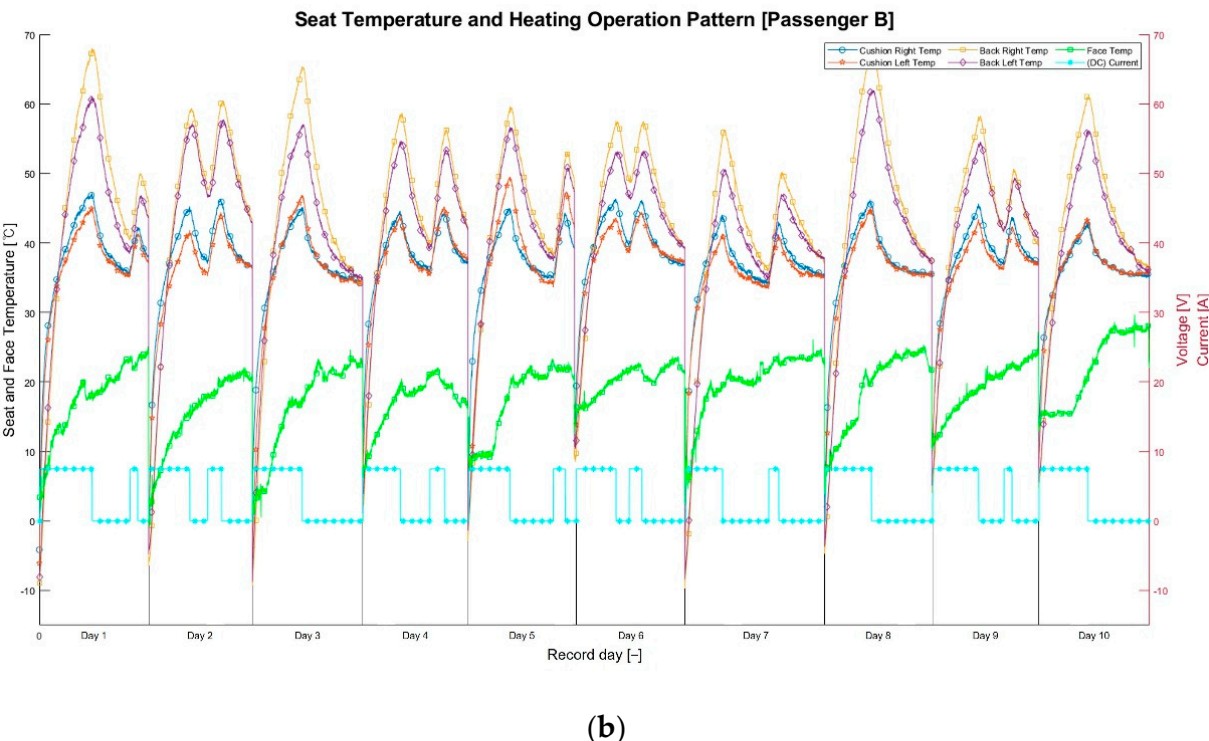

(**b**)

**Figure 5.** *Cont.*

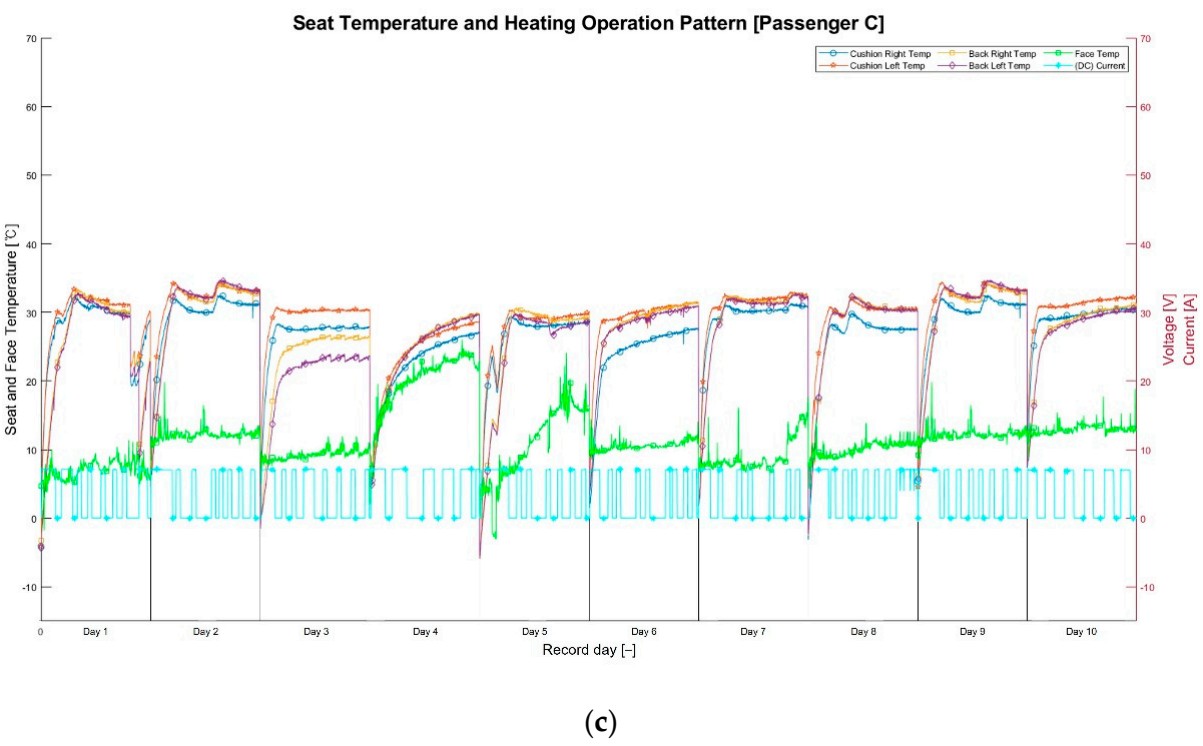

(**c**)

**Figure 5.** Seat temperature data of a vehicle: (**a**) passenger A; (**b**) passenger B; and (**c**) passenger C.

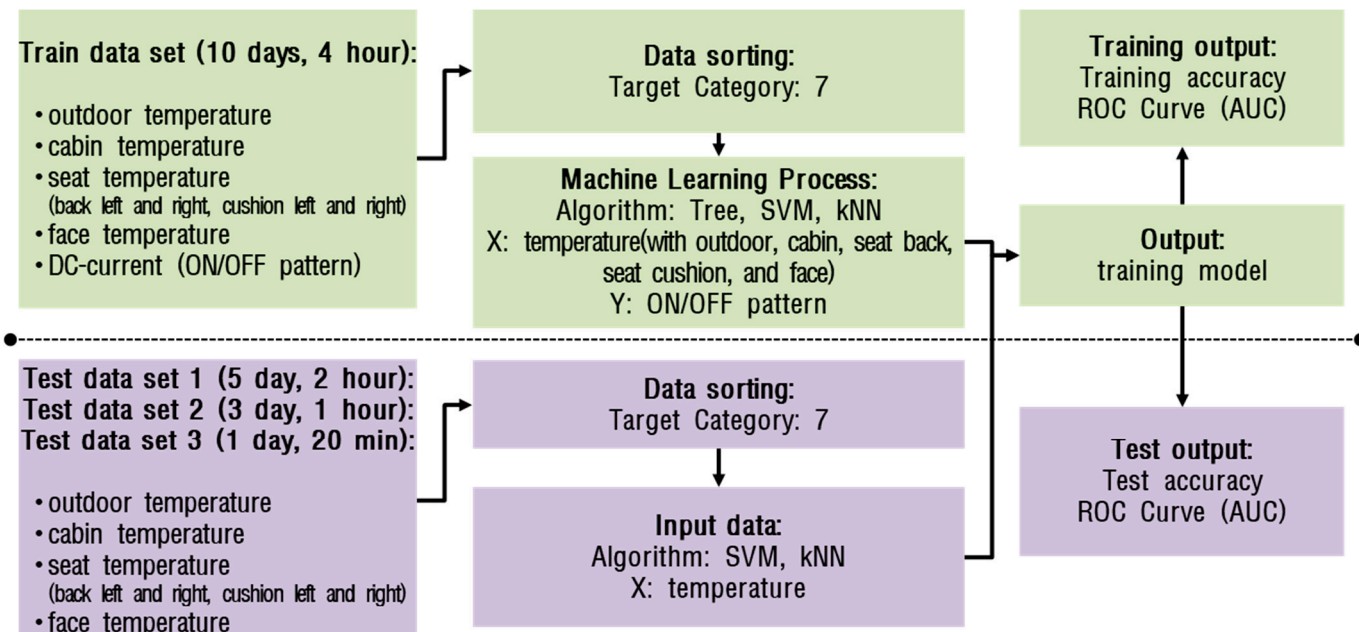

**Figure 6.** Input and output features used for personal thermal-comfort pattern model.

### 3. Machine Learning

*3.1. Machine-Learning Algorithm*

In this study, multiple classifications were made using a classification learner to create a personal thermal-comfort pattern model for each passenger. Valid data, measured for each passenger, was used in the training dataset for classification.

In addition, in the test dataset, the data corresponding to the recording time of approximately 2 h, approximately 1 h, and approximately 40 min (among the measurement data

excluded from the valid data) were recorded as test dataset 1, test dataset 2, and test dataset 3, respectively. They were used after converting to the input parameter of the personal thermal-comfort pattern model as the temperature (with outdoor, cabin, seat back, seat cushion, and face), and the output parameter was the ON/OFF pattern. Since the input parameter value-range was widely distributed for each passenger, the temperature data were converted using min–max normalization, except for the ON/OFF pattern for use in the classification learner. Min–max normalization is a method that linearly transforms original data and creates a balance between data before and after processing. Furthermore, it is used for machine learning [17,18]. The training and test datasets were transformed using min–max normalization. The classification training and testing were performed using MATLAB 2020a. The selected classification techniques were the support-vector machine (including linear, quadratic, cubic, and Gaussian kernels) and the k-nearest neighbor (including fine, medium, coarse, cosine, and cubic). Among the techniques widely used in public research, Tree, SVM, and kNN were employed, as they are reported to have high performance levels in classification learners [19,20].

The validation method of the machine-learning algorithm was 5-fold cross-validation. It divides the data into a fold, to avoid overfitting the dataset and estimate the accuracy of each fold. In the case of cross-validation, the trained model's two performance indicators used the training data. The validation account (VA) is the first metric used to evaluate the classification model, where 100% accuracy is perfect. The second indicator is the area under the receiver operating characteristic curve, which shows the probability of the true positive rate for the trained classification model. The area under the curve (AUC) is a single measure for estimating the positive acuity of a classification model. The AUC ranges from 0 to 1. An AUC of 1 indicates perfect accuracy, and an AUC of 0.5 indicates random guesses. With classification learners, the overall performance of the classification model was calculated by averaging the AUCs of all classes. In addition, the predictive performance can be confirmed through the true-positive rate (TPR) and false-positive rate (FPR).

The personal thermal-comfort pattern was predicted using the process shown in Figure 6. Using the dataset for training, the input parameters were divided into seven categories and the response was set as an ON/OFF pattern. The training using Tree, SVM, and kNN algorithms were performed. The algorithm for the test was determined by comparing the validation accuracy of the trained model with the AUC. Using the determined SVM and kNN algorithms, three test datasets were set as input parameters to perform the test. The personal thermal-comfort pattern was predicted using the accuracy and AUC values predicted from the test results.

*3.2. Results Using a Machine-Learning Algorithm*

Since the purpose of the training model using machine learning is to predict personal thermal comfort using the seat-heating operation pattern, the class of the model is well separated and predicted. As the training result of the predictive model for the data using the classification learner, the training model was derived after learning the heating ON/OFF pattern based on passenger action. Moreover, we verified whether personal thermal comfort is possible without a switch operation. The algorithm for the test was determined based on the validation accuracy (training accuracy) and AUC (predictive accuracy) according to the passenger. Table 3 summarizes the training accuracy results obtained from the training data. In the algorithm-training results for passenger A, kNN shows a higher training accuracy than Tree and SMV. The Tree and SVM show approximately 78% to 86% training accuracy, and kNN shows more than 90% training accuracy. For passenger B, Tree, SMV, and kNN, the training accuracy shows a high accuracy of over 91%. On the other hand, in the result for passenger C, the training accuracy ranges from a minimum of 63.9% to a maximum of 96% in the SVM. Compared to passengers A and B, the deviation of the training accuracy is approximately 30–40%. It was confirmed that the deviation of the training accuracy was up to 30%, even in Tree and kNN. It could be predicted that the heating-operation pattern, given the temperature of passenger C, was irregular compared to passengers A

and B. Through the training of Tree, SVM, and kNN, it was confirmed that the training in case of passenger B was the highest as a result of the accuracy compared to the passenger. Among the three algorithms for the passengers, the Tree algorithm had a lower average training accuracy than the other algorithms. In addition, as shown in Table 4, the AUCs of the Tree algorithm are 0.83, 0.95, and 0.76, which show a low predictive accuracy compared to the other algorithms. The average AUC deviation of the Tree algorithm and the SVM algorithm was similar, at approximately 3%. However, when the AUC of each passenger was compared, the SVM showed a high predictive accuracy. TPR and FRP also showed a 2% deviation from the SVM algorithm; however, the AUC difference was confirmed to be low, with a maximum deviation of 6%. Therefore, the test algorithm to predict each passenger's thermal-comfort pattern was determined using SVM and kNN.

**Table 3.** Training accuracy of all passengers.

| | Algorithm | Passenger A | Passenger B | Passenger C |
|---|---|---|---|---|
| Tree | Fine | 89.7 | 98.4 | 80.3 |
| | Medium | 82.7 | 91.7 | 72.1 |
| | Coarse | 77.9 | 86.8 | 63.9 |
| | Average | 83.4 | 92.3 | 72.1 |
| SVM | Linear | 77.3 | 92.4 | 72.3 |
| | Quadratic | 85.9 | 96.3 | 76.4 |
| | Cubic | 87.5 | 98.6 | 72.0 |
| | Fine Gaussian | 95.6 | 99.0 | 87.1 |
| | Medium Gaussian | 82.3 | 95.4 | 78.0 |
| | Coarse Gaussian | 78.2 | 91.7 | 72.2 |
| | Average | 84.5 | 95.6 | 76.3 |
| kNN | Fine | 97.6 | 99.7 | 90.4 |
| | Medium | 96.2 | 99.2 | 88.8 |
| | Coarse | 80.0 | 93.8 | 66.4 |
| | Cosine | 95.9 | 99.1 | 87.6 |
| | Cubic | 96.0 | 99.1 | 87.8 |
| | Weight | 97.5 | 99.6 | 90.4 |
| | Average | 93.9 | 98.4 | 85.2 |

**Table 4.** Training AUC of all passengers.

| | Algorithm | Passenger A | Passenger B | Passenger C |
|---|---|---|---|---|
| Tree | Fine | 0.92 | 0.98 | 0.86 |
| | Medium | 0.83 | 0.95 | 0.76 |
| | Coarse | 0.73 | 0.92 | 0.67 |
| | Average | 0.83 | 0.95 | 0.76 |
| SVM | Linear | 0.84 | 0.97 | 0.76 |
| | Quadratic | 0.88 | 0.98 | 0.72 |
| | Cubic | 0.82 | 0.98 | 0.73 |
| | Fine Gaussian | 0.98 | 0.98 | 0.94 |
| | Medium Gaussian | 0.90 | 0.98 | 0.73 |
| | Coarse Gaussian | 0.85 | 0.96 | 0.73 |
| | Average | 0.88 | 0.98 | 0.77 |
| kNN | Fine | 0.97 | 0.97 | 0.90 |
| | Medium | 0.97 | 0.98 | 0.96 |
| | Coarse | 0.91 | 0.98 | 0.74 |
| | Cosine | 0.98 | 0.97 | 0.95 |
| | Cubic | 0.98 | 0.98 | 0.95 |
| | Weight | 0.98 | 0.98 | 0.92 |
| | Average | 0.97 | 0.98 | 0.90 |

The test accuracy and AUC values were confirmed using the test dataset for the training models of SVM and kNN. Table 5 shows the test accuracy results for each passenger according to the test dataset. As shown in Table 5, in the test result of passenger A, the training-to-training accuracy increases in the two algorithms, SVM and kNN, and increases to a maximum of 5% in kNN. Both accuracies increase in the test dataset 1, and kNN (average VA = 96.8%) shows a higher average accuracy than SVM (average VA = 84.5%). In test dataset 2, the performances of SVM (average VA = 82.2%) and kNN (average VA = 73.7%), which are opposite to those of dataset 1, are shown to reduce by up to 23%. Test dataset 3 also shows a performance opposite to SVM (average VA = 80.0%) and kNN (average VA = 75.0%) to dataset 1, and reduces by up to 25%. The focus was confirmed in test datasets 2 and 3, and it was reduced by at least 15% to a maximum of 23% compared to dataset 1. In passenger B, the training-to-training accuracy decreases by up to 27% in the SVM algorithm, and the kNN algorithm also decreases by approximately 26%. In test dataset 1, the kNN algorithm decreases by up to 36%, and the average accuracy is higher in SVM (average VA = 75.4%) compared to kNN (average VA = 66.2%). Test dataset 2 shows similar SVM (average VA = 83.0%) and kNN (average VA = 76.8%) performance to dataset 1; however, it reduces by up to 27% in the SVM algorithm. In test dataset 3, kNN (average VA = 58.2%) shows a higher performance than SVM (average VA = 55.9%); however, it reduces by up to 43%. In test dataset 3, the performance is opposite to that of datasets 1 and 2, and it can be confirmed that it reduces by up to 32%. In addition, it can be confirmed that the test accuracy is low at 58%, and the learning is not well accomplished. In the case of passenger C, the accuracy compared to the training decreases because of a similar deviation to passenger B, and it decreases by up to 42% in the SVM algorithm and by up to 29% in the kNN algorithm. In test dataset 1, SVM decreases by up to 37%, and kNN (average VA = 67.1%) shows a higher average accuracy than SVM (average VA = 65.5%). In test dataset 2, SVM (average VA = 62.5%) and kNN (average VA = 64.5%), similar to those in dataset 1, are shown, and the SVM decreases by up to 31%. In test dataset 3, kNN (average VA = 66.5%) shows a high performance compared to SVM (average VA = 60.4%); however, it decreases by up to 43%. Test dataset 3 also shows a similar performance to dataset 2; however, it can be confirmed that it reduces by up to 42% in SVM. In addition, the average test accuracy is 62% in all test datasets, which shows a lower performance compared to passengers A and B.

**Table 5.** Test accuracy of all passengers.

|  | Algorithm | Passenger A | | | Passenger B | | | Passenger C | | |
|---|---|---|---|---|---|---|---|---|---|---|
|  |  | Dataset 1 | Dataset 2 | Dataset 3 | Dataset 1 | Dataset 2 | Dataset 3 | Dataset 1 | Dataset 2 | Dataset 3 |
| SVM | Linear | 81.1 | 83.8 | 80.6 | 74.0 | 87.8 | 55.5 | 77.4 | 69.0 | 67.0 |
|  | Quadratic | 84.4 | 90.9 | 84.5 | 82.1 | 87.6 | 57.7 | 53.5 | 54.1 | 59.9 |
|  | Cubic | 78.9 | 75.8 | 71.3 | 68.0 | 78.5 | 58.4 | 72.5 | 59.7 | 53.3 |
|  | Fine Gaussian | 97.1 | 77.7 | 70.8 | 76.8 | 72.5 | 58.3 | 54.8 | 60.4 | 50.8 |
|  | Medium Gaussian | 86.2 | 74.9 | 86.0 | 81.3 | 82.3 | 53.3 | 57.4 | 64.0 | 64.5 |
|  | Coarse Gaussian | 79.2 | 80.3 | 86.5 | 79.0 | 89.2 | 52.4 | 76.8 | 67.7 | 66.8 |
|  | Average | 84.5 | 82.2 | 80.0 | 75.4 | 83.0 | 55.9 | 65.5 | 62.5 | 60.4 |
| kNN | Fine | 99.8 | 72.2 | 74.5 | 63.9 | 74.2 | 58.9 | 67.5 | 62.7 | 64.3 |
|  | Medium | 98.2 | 74.4 | 74.3 | 66.4 | 74.9 | 58.1 | 70.0 | 64.8 | 63.1 |
|  | Coarse | 86.7 | 71.0 | 78.1 | 72.8 | 81.1 | 57.3 | 65.7 | 67.2 | 74.5 |
|  | Cosine | 97.9 | 77.6 | 75.3 | 62.5 | 82.2 | 58.5 | 65.6 | 64.3 | 65.6 |
|  | Cubic | 98.1 | 73.6 | 72.0 | 66.3 | 73.7 | 57.9 | 67.3 | 62.1 | 63.3 |
|  | Weight | 99.8 | 73.5 | 75.5 | 65.5 | 74.4 | 58.4 | 66.6 | 66.0 | 62.4 |
|  | Average | 96.8 | 73.7 | 75.0 | 66.2 | 76.8 | 58.2 | 67.1 | 64.5 | 65.5 |

As shown in Table 6, according to the test data set, the AUC results tend to increase for passenger A; however, they decrease for passengers B and C. For passenger A, the overall increase in the SVM algorithm increases by up to 12% compared with the training AUC; however, there are cases where it decreases by up to 15%. In test dataset 1, AUC increases up to 9% in SVM (average AUC = 0.9), showing a high predictive accuracy compared with kNN. In kNN (average AUC = 0.95), AUC tends to decrease; however, with a slight deviation of approximately 2%. In test dataset 2, there is a 15% decrease in SVM (average AUC = 0.9); however, it increases overall. In addition, it an be confirmed that the kNN (average AUC = 0.96) decreases with a slight deviation of approximately 2%. In test dataset 3, the AUC increases from 1% to 2% in both SVM (average AUC = 0.90) and kNN (average AUC = 0.98), confirming that it is similar to the training AUC. Passenger B confirms that the AUC decreases, compared to training in both algorithms, and it can confirmed that it is the lowest in test dataset 3. In test dataset 1, AUC is higher in SVM than in kNN, and AUC reduces by up to 22% in SVM (average AUC = 0.84). In addition, in kNN (average AUC = 0.72), it reduces by up to 30%. In test dataset No. 2, SVM (average AUC = 0.86) and kNN (average AUC = 0.87) show a similar trend, and it can be confirmed that the decrease is by up to 16%. In test dataset 3, the AUC is the lowest in both SVM (average AUC = 0.6) and kNN (average AUC = 0.59), and it an be confirmed that the most significant deviation from training AUC occurs by decreasing up to 40%. In the case of passenger C, the AUC shows a similar trend according to the test dataset, and the AUC of kNN is approximately 5% higher than that of SVM. In test dataset 1, AUC decreases by up to 37%, compared to training AUC in SVM (average AUC = 0.64). The AUC decreases by up to 28% in kNN (average AUC = 0.69). In test dataset 2, similar to dataset 1, the AUC of kNN (average AUC = 0.66) is high and decreases by up to 28%. In addition, the SVM (average AUC = 0.59) reduces by up to 29%, and it can be confirmed to be the lowest among the AUCs. In test dataset 3, kNN (average AUC = 0.64) decreases by up to 32% and SVM (average AUC = 0.61) decreases by up to 43%—confirming that it decreases with the most significant deviation.

**Table 6.** Test AUC of all passengers.

|  | Algorithm | Passenger A | | | Passenger B | | | Passenger C | | |
|---|---|---|---|---|---|---|---|---|---|---|
|  |  | Dataset 1 | Dataset 2 | Dataset 3 | Dataset 1 | Dataset 2 | Dataset 3 | Dataset 1 | Dataset 2 | Dataset 3 |
| SVM | Linear | 0.85 | 0.92 | 0.84 | 0.85 | 0.89 | 0.59 | 0.67 | 0.60 | 0.66 |
|  | Quadratic | 0.91 | 0.94 | 0.88 | 0.89 | 0.89 | 0.66 | 0.62 | 0.52 | 0.64 |
|  | Cubic | 0.89 | 0.85 | 0.82 | 0.82 | 0.82 | 0.59 | 0.73 | 0.60 | 0.56 |
|  | Fine Gaussian | 0.99 | 0.83 | 0.99 | 0.76 | 0.86 | 0.60 | 0.59 | 0.72 | 0.54 |
|  | Medium Gaussian | 0.91 | 0.97 | 0.90 | 0.88 | 0.88 | 0.61 | 0.57 | 0.52 | 0.68 |
|  | Coarse Gaussian | 0.86 | 0.89 | 0.85 | 0.86 | 0.82 | 0.57 | 0.64 | 0.60 | 0.62 |
|  | Average | 0.90 | 0.90 | 0.90 | 0.84 | 0.86 | 0.60 | 0.64 | 0.59 | 0.61 |
| kNN | Fine | 0.97 | 0.98 | 0.89 | 0.68 | 0.89 | 0.59 | 0.66 | 0.36 | 0.61 |
|  | Medium | 0.96 | 0.94 | 0.99 | 0.73 | 0.83 | 0.58 | 0.70 | 0.68 | 0.65 |
|  | Coarse | 0.86 | 0.91 | 0.91 | 0.79 | 0.86 | 0.59 | 0.72 | 0.64 | 0.65 |
|  | Cosine | 0.95 | 0.97 | 0.99 | 0.67 | 0.87 | 0.57 | 0.69 | 0.63 | 0.68 |
|  | Cubic | 0.96 | 0.97 | 0.99 | 0.73 | 0.87 | 0.58 | 0.68 | 0.70 | 0.68 |
|  | Weight | 0.97 | 0.98 | 0.99 | 0.73 | 0.88 | 0.59 | 0.70 | 0.68 | 0.72 |
|  | Average | 0.95 | 0.96 | 0.98 | 0.72 | 0.87 | 0.59 | 0.69 | 0.66 | 0.64 |

The test results using the whole dataset confirm that the overall test accuracy and AUC decrease. The test accuracy generally decreases compared to the training accuracy according to the passenger; however, show a similar trend in the case of passenger A. As shown in Figure 7, the valid results for each passenger are confirmed according to the test dataset. Passenger A shows the highest accuracy when using dataset 1 among the test datasets. Passenger B shows high accuracy when using dataset 2 but shows the lowest

accuracy of 58% in the case of dataset 3. Passenger C shows a similar performance in the kNN algorithm according to the dataset; however, it can be confirmed that the accuracy is relatively low at 60%. In the case of AUC, as shown in Figure 8, it decreases by at least 8% to a maximum of 43%, compared to the training AUC for each passenger; however, the AUC increases in the case of passenger A or shows a similar trend. Confirming the valid results according to the test dataset, passenger A shows similar values for each dataset. When using datasets 2 and 3, the highest AUC (kNN TPR = 0.98; kNN FPR = 0.07) appears. Passenger B shows a high AUC (SVM TPR = 0.97; SVM FPR = 0.02) when using dataset 2, and shows a relatively low AUC in the case of dataset 3. Passenger C shows a similar trend in the dataset, and it can be confirmed that when dataset 1 is used, a relatively high AUC (kNN TPR = 0.71; kNN FPR = 0.39) emerges. It can be confirmed that passenger A's "learning" results show the highest predictive accuracy among the passengers. This shows that the cycle of passenger A's ON/OFF pattern data has a high correlation with the temperature data. Hence, the accuracy and AUC tends to be high, and a slight deviation from the training result appears.

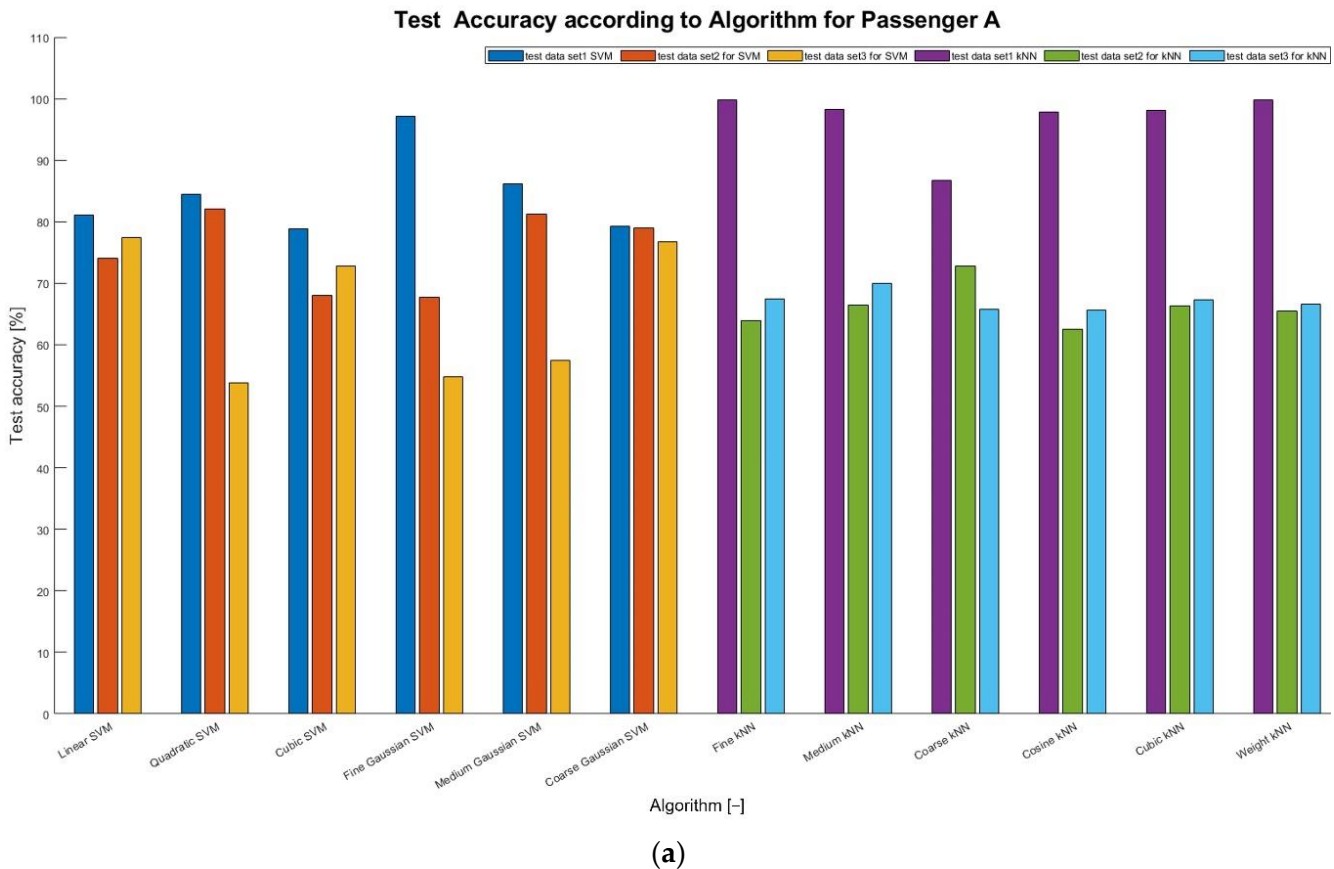

(**a**)

**Figure 7.** *Cont.*

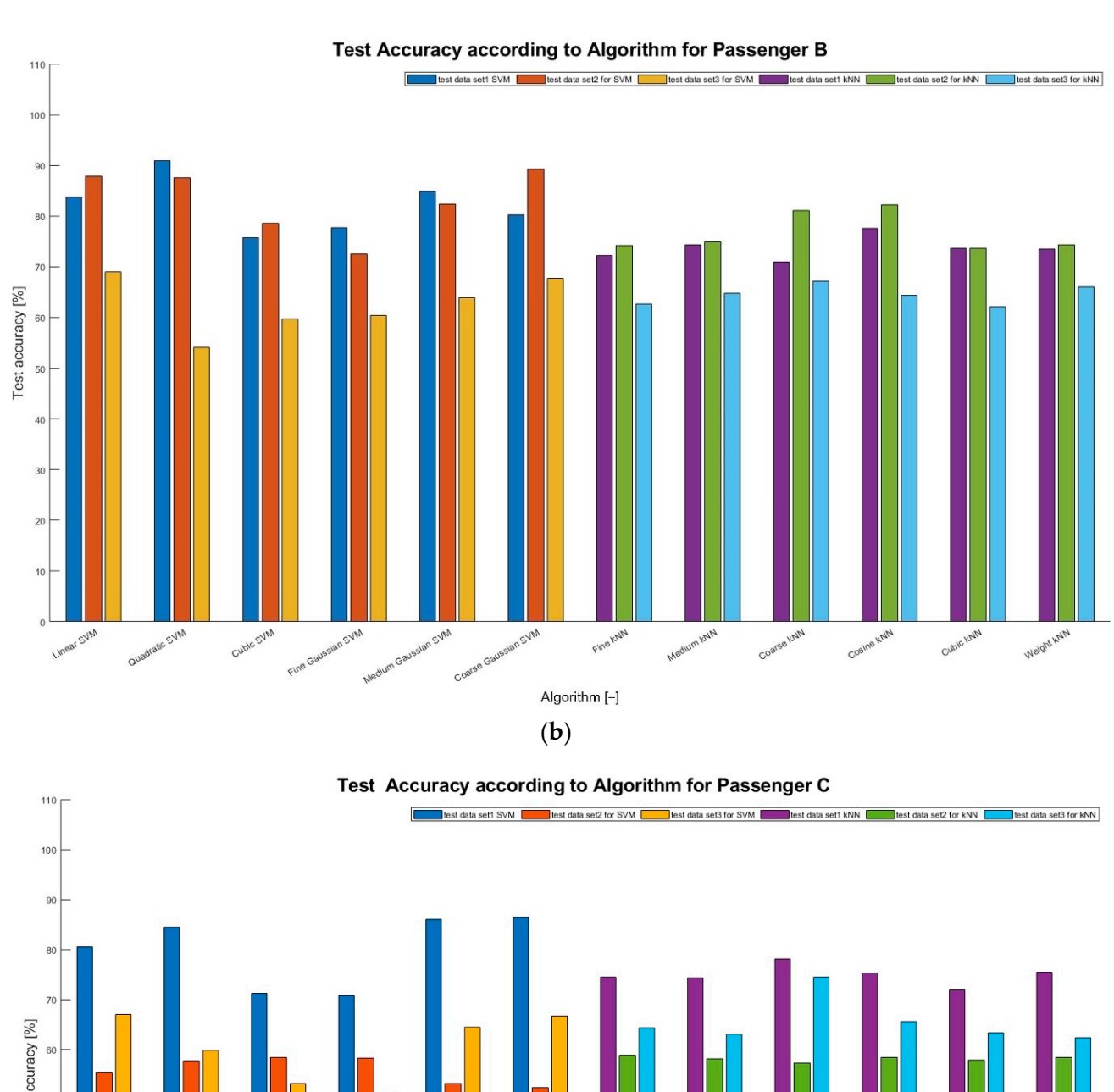

**Figure 7.** Test accuracy of all the passengers: (**a**) test accuracy of passenger A; (**b**) test accuracy of passenger B; and (**c**) test accuracy of passenger C.

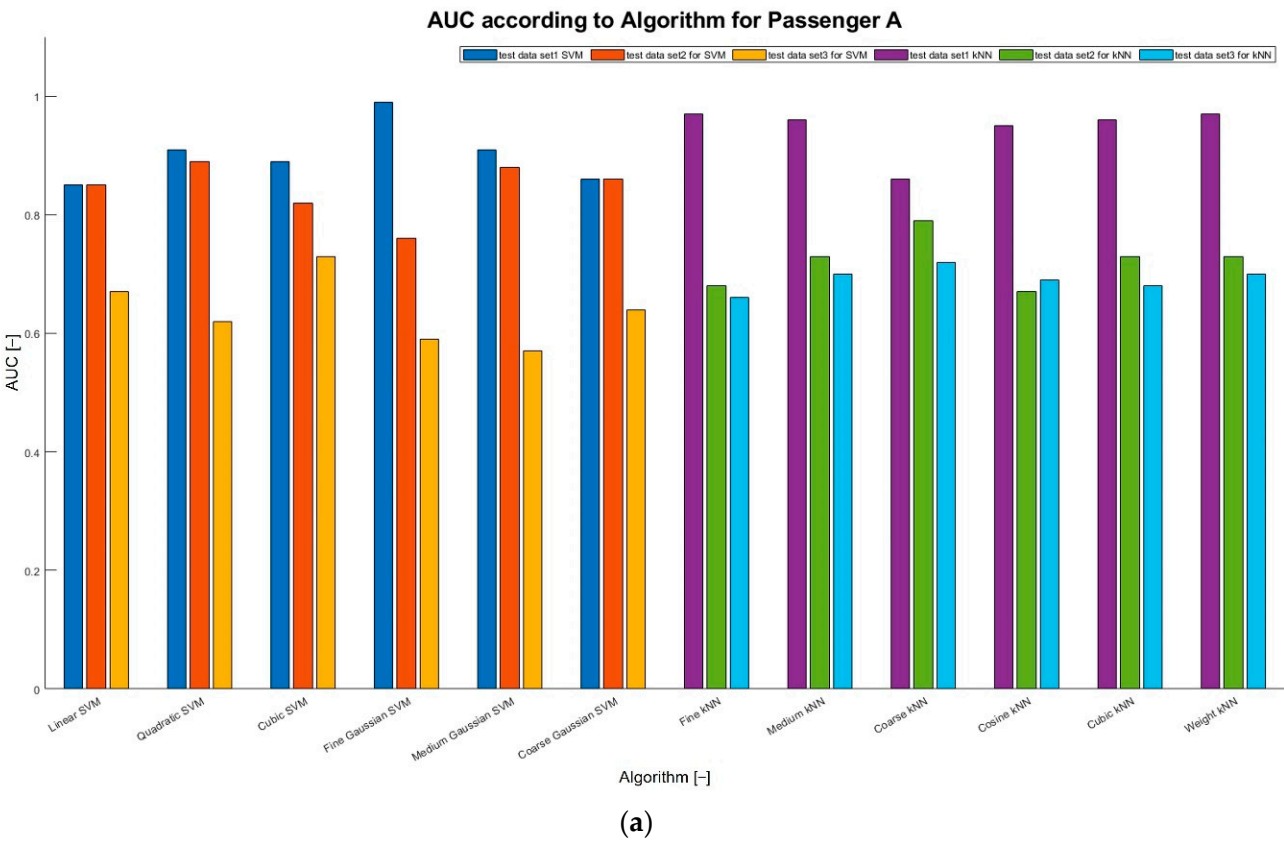

(**a**)

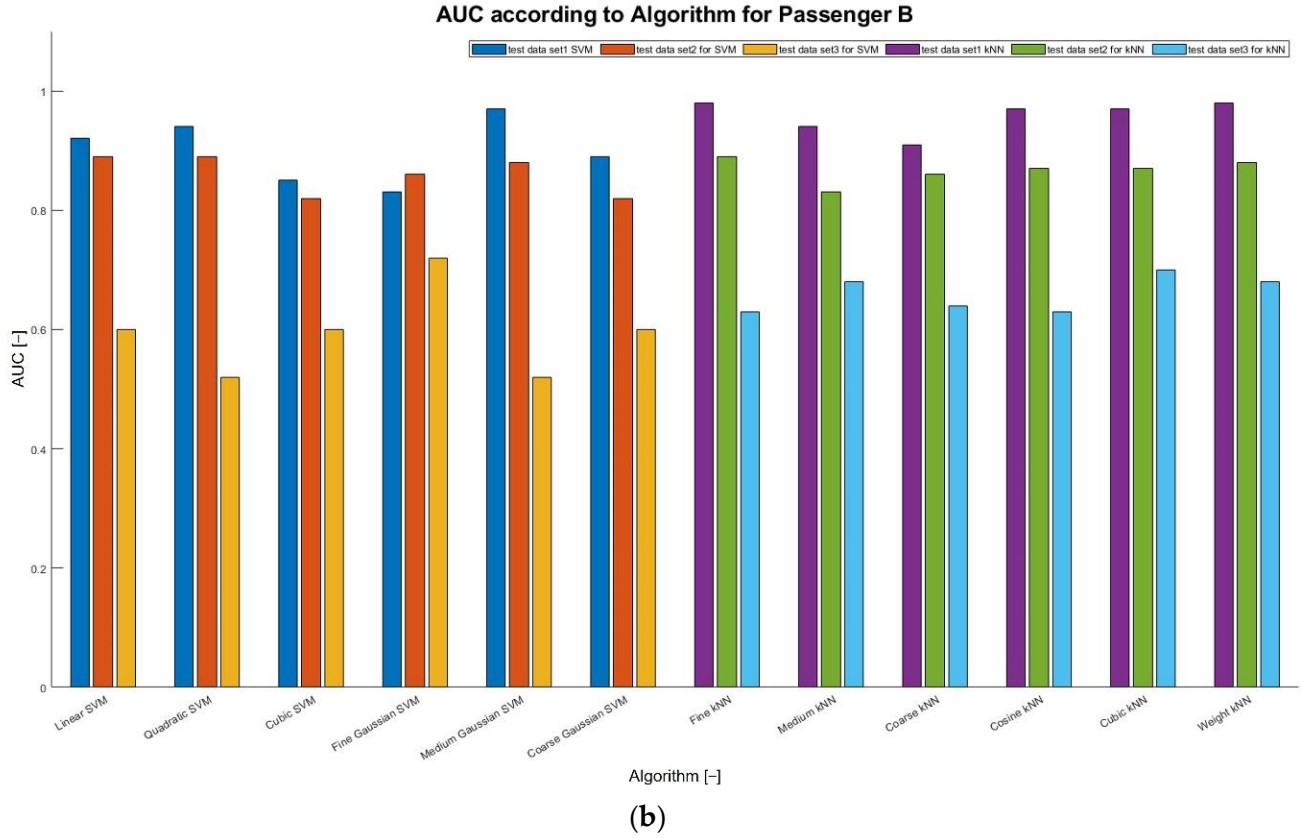

(**b**)

**Figure 8.** *Cont.*

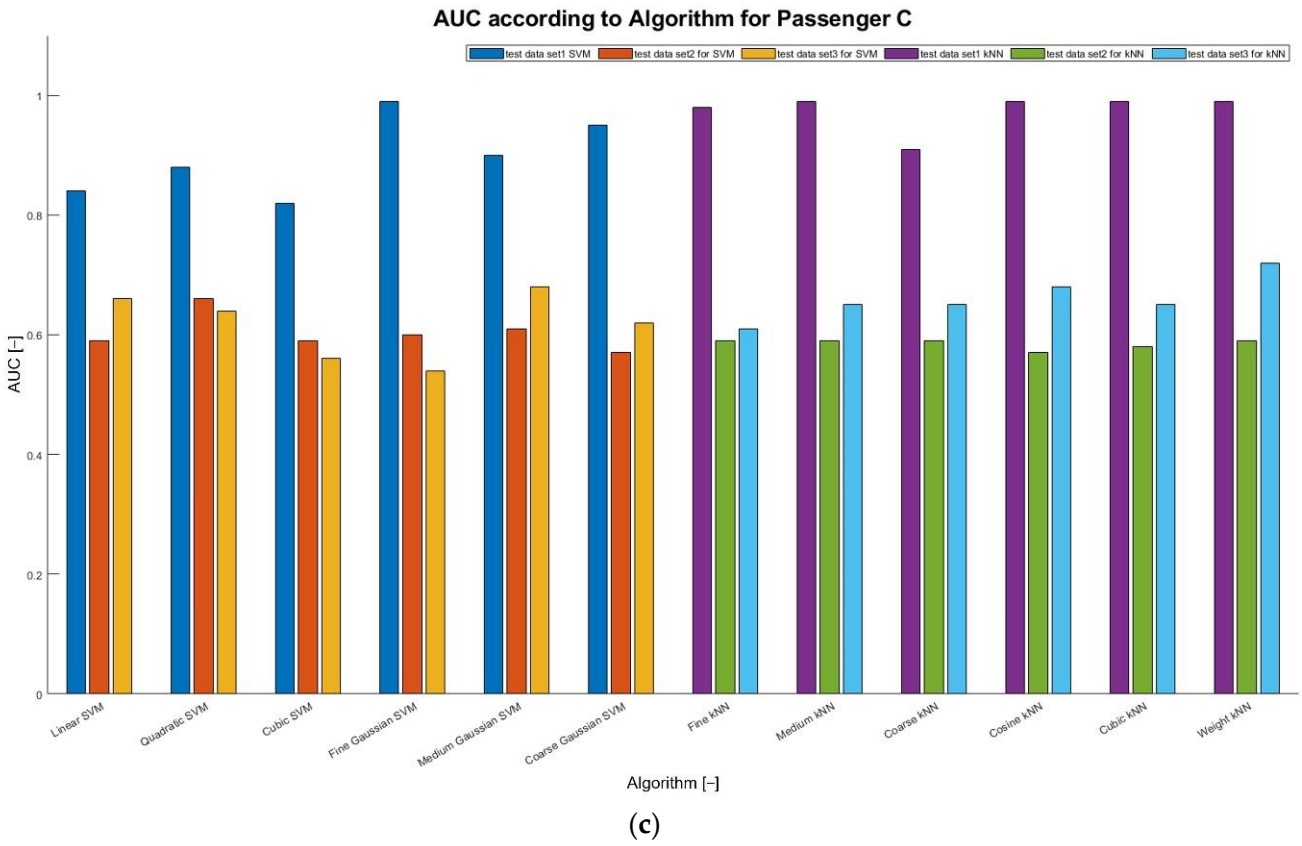

**Figure 8.** AUC of all passengers: (**a**) AUC of passenger A; (**b**) AUC of passenger B; and (**c**) AUC of passenger C.

## 4. Discussion

Section 3 estimates the appropriate dataset and algorithm for the prediction result according to the seat-heating ON/OFF pattern. That was achieved by using machine-learning training. In the accuracy result of the seat-heating ON/OFF pattern based on passenger action, passenger A showed a high accuracy in kNN using dataset 1. For passenger B, the SVM using dataset 2 showed a high accuracy. In addition, passenger C showed a low accuracy overall; however, SVM using dataset 1 showed a relatively high accuracy. In the AUC results, passenger A showed a high AUC with kNN using datasets 2 and 3, and passenger B showed a high AUC with SVM using datasets 1 and 2. kNN using dataset 2 also showed a high performance.

In addition, passenger C ranked lower than other passengers; however, when datasets 1 and 2 were used, kNN showed a relatively high AUC. Accordingly, it was confirmed that a relatively high prediction was possible using test datasets 1 and 2 with the kNN algorithm. In the prediction results, the deviation between test datasets 1 and 2 was small. Accordingly, it was shown that learning was possible, even when using a dataset corresponding to approximately 30% of the training dataset.

When using the test datasets, as shown in Figure 9, it can be confirmed that the average predictive accuracy ranges from 60% to 96%. In addition, TPR predicting the true value shows a minimum of 0.71 and a maximum of 0.98, while FPR predicting a false value shows a minimum of 0.02 and a maximum of 0.39. Although the predictive accuracy of passenger C had a large deviation compared to other passengers, it was confirmed that the heating pattern could be predicted. This can be expected because the passenger is not sensitive to temperature. Therefore, the heating pattern is not regular.

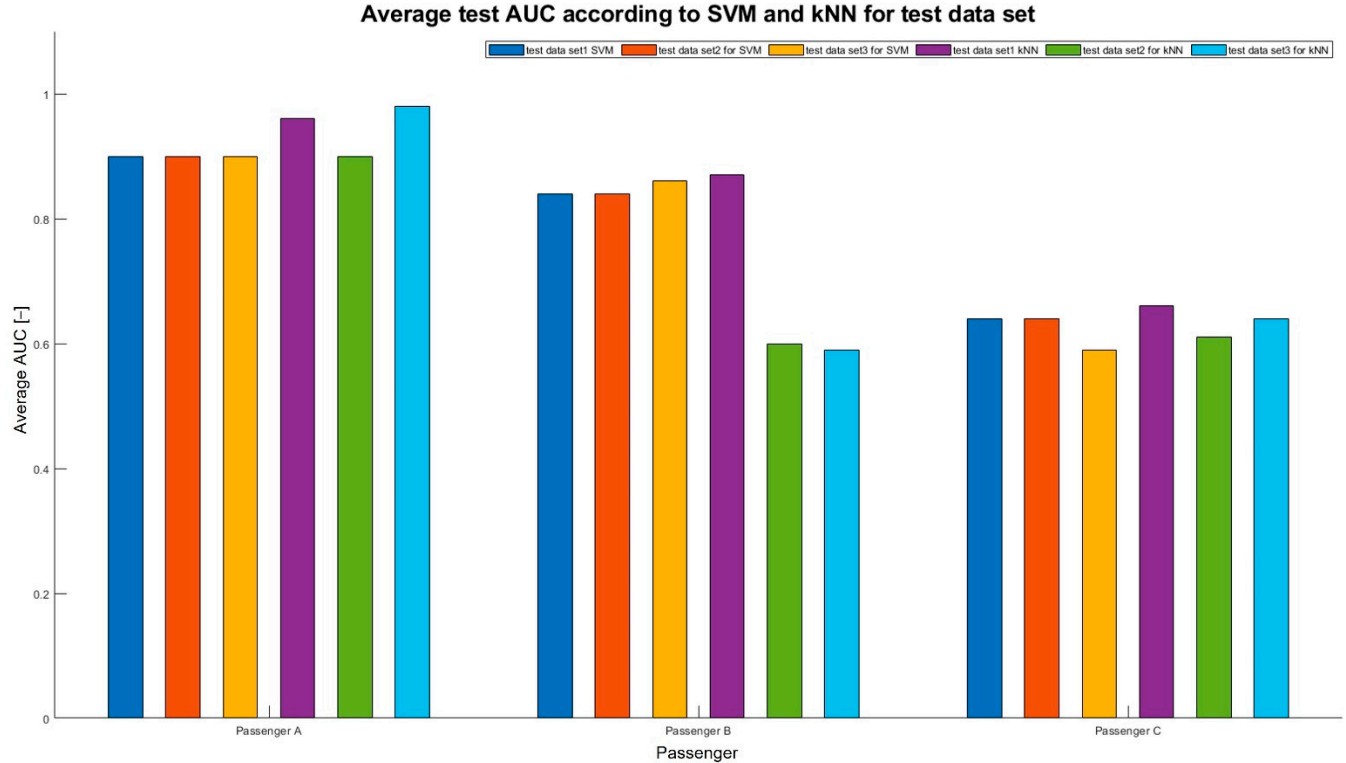

**Figure 9.** Average test AUC of all the passengers.

It was found that learning-based prediction was possible when the train was "trained" in a winter environment condition, in addition to the experimental environment. This was despite a deviation in the heating pattern that satisfied the personal thermal comfort, depending on the passenger. Accordingly, it was found that the ON/OFF of the heating seat was possible without the passenger operating the switch.

As a prediction result, we discuss why the approach using the ON/OFF pattern was tested and how the application of this learning model is reflected. For thermal comfort, various scales are widely used to evaluate individual experiences with conditions—most commonly the thermal-sensation scale, also known as the thermal-comfort scale [21]. It is suitable for describing the one-dimensional relationship between the physical parameters (of an indoor or cabin environment) and the subjective thermal sensation.

However, the thermal comfort of an occupant or passenger is a complex phenomenon, depending on the psychological and physiological factors. It is not easy to measure. The passenger's thermal comfort may also disagree with the range of comfort scales classified as comfortable. Cabin environments, such as cars (not indoors), may have different measures of thermal comfort for each individual caused by various parameters in the "narrow environment". Therefore, thermal comfort is mainly predicted using the relationship between the ambient temperature and skin temperature.

Due to individual differences, personal thermal comfort has to predict individual needs, not a generalization of people's needs. If perceived differently on behalf of each passenger, it is possible to improve a satisfactory preference for thermal comfort by reflecting the individual's direct feedback.

Conversely, it is possible to reduce the rate of thermal dissatisfaction expressed by individual passengers in the cabin [22]. The heating seat operation pattern performed in this study reflects an individual's direct feedback. It can be said to be a similar approach to the thermal-comfort vote. This approach applied to the actual winter environment shows that the passenger directly responds to the outdoor, cabin, and seat temperatures. It helps to learn and predict the operation pattern of the heating seat to achieve personal thermal comfort without operating the heating switch.

In a similar study, a thermal-sensation vote was used to predict the thermal comfort through machine learning. In the study, the possibility of its application in other environments was confirmed by learning patterns.

Machine-learning algorithms and data-pattern approaches provide flexibility for testing different modeling methods and potential input variables affecting thermal comfort. New input variables related to personal thermal comfort can be introduced through a model based on pattern data. A heating pattern in another vehicle, a ventilation operation pattern in a summer environment, or an adjustment pattern of a seat may be considered as a new input variable.

The focus of this study was to use the outdoor temperature, cabin temperature, seat temperature, and face-skin temperature as the inputs for the heating-operation pattern. From the perspective of using a heating seat in an actual vehicle environment, the focus recognizes the passenger and ambient temperature of the front row seat. It predicts it through machine learning to control the temperature of the seat without operating the heating switch. Furthermore, it can be applied to ventilation and heating, and the method of this study can also be applied to the seat air-conditioning system of a passenger in the rear row. The additional variables to consider here are the overall HVAC system of the cabin, the clothes of the passenger, or the health of the passenger. In addition, the machine-learning algorithm used can be a variable, affecting pattern prediction.

## 5. Conclusions

As each passenger has different comfort requirements in the car-cabin environment, personal thermal comfort is perceived differently. Therefore, the focus of this study was to investigate the predictability of personal thermal comfort using machine-learning methods. We predicted various passenger operation patterns for heating seats in vehicles in a winter driving environment. The combination of input and output variables was tested using a classification learner with the operation-pattern data of the heating seat.

For the data, we measured the pattern corresponding to outdoor and cabin temperatures, while the passengers used MFMSD. They drove about 310 km in winter. Train and test datasets were created by setting the measured data as input and output variables, using min–max normalization for machine learning.

The AUCs of 0.83, 0.87, and 0.95 were generated by three algorithms, Tree, SVM, and kNN, respectively, which was a result of training according to the input and output variables. As a result of the test using SVM and kNN, excluding the Tree with a relatively low train AUC, passenger A had the highest AUC value of 0.89, and passenger C showed a low AUC value of 0.62. In addition, the test results based on the test dataset generally showed an AUC value of 0.75 or more—indicating that the passenger's pattern prediction value was high. These results show that the heating-operation pattern for predicting personal thermal comfort can be recognized.

The results of this study are expected to be used as preliminary research on temperature control according to the passenger in a winter driving environment. In addition, this study is expected to be utilized to construct an AI-based HVAC system that controls the passenger's car-seat temperature through machine learning.

**Author Contributions:** Conceptualization, E.S.J. and J.R.L.; methodology, E.S.J., J.R.L. and Y.J.J.; software, E.S.J. and Y.J.J.; validation, E.S.J., J.R.L. and Y.J.J.; formal analysis, E.S.J. and J.R.L.; resources, E.S.J., J.R.L. and Y.J.J.; data curation, J.R.L. and Y.J.J.; writing—original draft preparation, E.S.J. and J.R.L.; writing—review and editing, E.S.J., J.R.L. and Y.J.J.; visualization, E.S.J. and Y.J.J.; supervision, E.S.J. and J.R.L.; project administration, E.S.J. and Y.J.J.; funding acquisition, E.S.J. and J.R.L. All authors have read and agreed to the published version of the manuscript.

**Funding:** This work was supported by the research grant of the Kongju National University in 2021.

**Informed Consent Statement:** Informed consent was obtained from all subjects involved in the study.

**Data Availability Statement:** Data is contained within the article.

**Conflicts of Interest:** The authors declare no conflict of interest.

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
