# Peer review of "Prediction of AI-Based Personal Thermal Comfort in a Car Using Machine-Learning Algorithm"

_electronics, doi:10.3390/electronics11030340_

Round 1

Reviewer 1 Report

The paper proposes an artificial intelligence-based system to control the seat heating temperature for passengers through machine learning. I find the paper very interesting and well-written. However, the authors need to address the following issues:
1. In the very first paragraph of "Introduction", the justification behind the statement "predicting passengers' thermal comfort in a vehicle is an important issue" is not very convincing. This is the prime motivation for this work and should be justified with better evidence. There are several machine learning-based related works, as the authors have pointed out, however, the novelty of the contributions of this paper needs to be highlighted with a proper gap analysis with existing literature.

2. Why KNN and SVM algorithms are chosen over other ML algorithms needs to be clarified.

3. The sensor data is only for the driver, but will it be equally applicable to all other passengers of the cars? If possible, add more variation in the data.

4. Please set propoer colors to the bars in Fig. 10(a). As you have 12 bars for each passenger, the MATLAB plot is repeating the same colors after 7 bars and it becomes confusing to follow the legend. Also, increase the font-size of the labels/legends in all the figures and use vector images to improve the quality of the figures.

5. The description for the results are very trivial and summarizes the accuracy of the algorithms only. However, no insights were deduced from the results that can highlight the merits of the proposed model by the authors.

Reviewer 2 Report

First, the word "control" in the title is misleading: this paper is not about controlling anything. It is about prediction something called “passenger's temperature control sensibility” or “thermal control sensibility “, a concept which is not defined.

There are no clear statements about the objective of the study, what problem it tries to solve, and why the proposed solution is better with respect to the state of the art.

Some phrases do not make much sense; e.g. in lines 42-43 "The heating or ventilation performance while a passenger  is sitting on it is a significant factor for thermal comfort” , and in lines  201-202 “the data recorded when the ON/OFF operation of the heating seat was not performed while driving were excluded”.

Some references are cited incorrectly e.g ref [7] is mentioned in the text as "Alok et al." instead of "Warey et al."

Threfore, I believe that the presentation needs MAJOR revision before any serious evaluation of the contributions of this study.

Round 2

Reviewer 1 Report

I have no further comments. Thanks.

Author Response

Dear Editor of Electronics:

Thank you very much for your comment.

Thank you for your review.

Reviewer 2 Report

This version of the paper has been almost entirely rewritten, compared with the initial version. The authors addressed all the suggestions I made in the first round of review, and I will not formulate other critics (except this minor suggestion: please select the keywords so that they describe more specifically the content of the paper. Keywords like "prediction", "machine learning algorithm:, "in a car" are much too general.)
